# Use of Visual Pedagogy to Help Children with ASDs Facing the First Dental Examination: A Randomized Controlled Trial

**DOI:** 10.3390/children9050729

**Published:** 2022-05-16

**Authors:** Silvia Cirio, Claudia Salerno, Stefania Mbanefo, Luca Oberti, Lujbicca Paniura, Guglielmo Campus, Maria Grazia Cagetti

**Affiliations:** 1Department of Biomedical, Surgical and Dental Science, University of Milan, Via Beldiletto 1, 20142 Milan, Italy; silviacirio@alice.it (S.C.); claudia.salerno@unimi.it (C.S.); stefania.mbanefo@unimi.it (S.M.); luca.oberti1@unimi.it (L.O.); ljubicca.paniura@unimi.it (L.P.); 2Department of Medicine, Surgery and Experimental Sciences, School of Dentistry, University of Sassari, Viale San Pietro, 07100 Sassari, Italy; guglielmo.campus@zmk.unibe.ch; 3Department of Restorative, Preventive and Pediatric Dentistry, University of Bern, Freiburgstrasse 7, 3010 Bern, Switzerland

**Keywords:** ASDs children, dental visit, visual pedagogy

## Abstract

Autism Spectrum Disorders (ASDs) are neurodevelopmental disorders that don’t have a direct effect on oral health, but severe difficulties in oral hygiene and dental procedures expose people with ASDs to an increased risk of oral diseases. This RCT aimed to evaluate which pedagogical tool was the best to prepare children with ASDs for their first dental examination, either video or photo aids. Two different criteria were used to evaluate their efficacy: the achieved steps into which the procedure was divided (*n* = 8), and the level of cooperation according to the Frankl Behavioral Scale. One hundred-thirteen subjects were randomly assigned to the two groups and 84 subjects completed the trial (Video group *n* = 41; Photo group *n* = 43). A predictive model for the achievement of the Preliminary (1–4) or Dental (4–8) steps was performed using a multivariate logistic regression procedure. Children in the Video group achieved more steps, but the comparison between groups was statistically significant only for the Preliminary steps (*p* = 0.04). The percentage of subjects judged as cooperating was similar in the two groups. The results of this study underline that behavioural intervention should be used as an effective strategy to prepare subjects with ASDs for a dental examination.

## 1. Introduction

Autism Spectrum Disorders (ASDs) are neurodevelopmental disorders [1] defined as “A complex biological disorder that generally lasts for a person’s entire life, beginning before the age of three, in the developmental period, and causes delays or problems in many different ways in which a person develops or grows” [2].

ASDs do not have a direct oral health outcome, but they are associated with severe difficulties in dental procedures at home and in office, exposing people with ASDs to an increased risk of oral diseases, such as caries lesions, gingivitis, periodontal disease and traumatic injuries [3]. Specific strategies to maintain oral health are necessary in subjects with ASDs, as they can occur in a broad range of clinical conditions and severity. The poor social interaction and communication skills of these subjects are the first obstacle that the dentist needs to overcome in their management and treatment. Moreover, compromised communication skills have been associated with behavioural problems [4]. The subjects with ASDs have greater difficulty in managing their emotions, as they may exhibit stereotyped movements, low frustration tolerance, and hyperactivity with attention deficit that may invalidate their behaviour in the dental chair [5]. Sensory hypersensitivity is another feature to take into consideration by dentists [6]. Furthermore, the dental environment is unusually unfamiliar compared to the school environment, which often causes defiant behaviour [7]. All of these characteristics can lead to an unsuccessful dental treatment, which is often managed with sedative drugs or dental treatments under general anaesthesia [8,9,10].

Parents and therapists are an essential and valuable aid in improving collaboration with patients and reducing negative and oppositional behaviours during dental care [11]. Before the dental visit, an interview with the parents/caregiver is essential to investigate behavioural problems, preferences, individual characteristics and the needs of the child [12]. This information makes it possible to select the most effective behavioural approach to dental care and setting. As each patient reacts differently to different behavioural strategies, these must be adapted to each child [13,14].

Various behavioural techniques are described; the most common of the counter-conditioning procedures are systematic desensitization, the “tell-show-do” technique, voice control, positive reinforcement, distraction and visual pedagogy [5,14,15]. Visual pedagogy, together with positive reinforcement, is a cognitive behavioural therapy commonly used to treat subjects with ASDs. Visual aids can be photographic images, drawings, symbols, or videos; in some cases, they can be combined with short texts as in social stories, in paper or digital format and organized as different types of pedagogical tools [16,17]. Social stories are short stories of visual elements such as images, drawings or photos associated with descriptive sentences [18]. Social stories can help patients with ASDs to understand what will happen during the visit or the dental procedures [5]. Video modelling consists of watching a video that shows a subject engaged in the skill that needs to be improved [19]. Although limited evidence exists on the use of video modelling in dentistry [20,21], practitioners can find videos specifically realized to improve home oral hygiene procedures and compliance during dental examination.

Based on this premise, a randomized controlled trial was designed, planned and performed to evaluate the effectiveness of two visual aids in preparing children with ASDs for their first dental examination at the dental Clinic of University of Milan, Italy, where a behavioural strategy based on visual aids had previously been developed [16].

## 2. Materials and Methods

### 2.1. Study Design

This randomized controlled clinical trial was conducted between December 2020 and January 2022. Parents of children with ASDs, aged between three and 14 years, who phoned the dental clinic asking for an examination, were invited to participate. During the call, the purpose of the study was explained and, after a verbal consent to participate, a written consent form based on the Declaration of Helsinki was sent via email, asking parents to return it completed and signed (Ethics Committee of San Paolo Hospital N 273).

### 2.2. Procedures and Evaluation

The primary outcome was the comparison of the performances of the two visual aids in increasing the child’s cooperation during the first dental examination. Two visual aids were prepared. A video recorded in the dental office with a cooperative patient as a model was prepared for the Video group (*Vg*) (Appendix A). The parents were informed of the purpose of the recording and signed consent. The video was edited by inserting a relaxing musical theme and removing background noise to avoid distraction of the child during the projection. In the video, the patient and the dentists reproduced all the actions usually performed during the dental visit with an external narrator voice. The environment, dentists and tools shown were the same as those used during the dental visit. For the Photo group (*Pg*), a social story was prepared (Appendix A), in which the first dental visit was described with photographic images and short sentences. Again, the environment, dental personnel, and instruments were the same as those used during the dental visit. The visual aids were randomized and delivered to the parents of the children via email at least three weeks before the appointment.

Two different criteria were used to evaluate the effectiveness of the two visual aids:Evaluation of the steps, derived from similar studies [22,23,24], needed to complete a first dental examination:The patient enters the room;The patient sits in the dental chair;The patient leans with his back against the dental chair and lets the operator recline at least 45°;The patient accepts that the operator turns on the light of the dental chair and points it at the mouth area;The patient opens his/her mouth;The patient allows the operator to count the teeth with his fingers;The patient accepts the inspection of the mouth with a dental mirror;The patient accepts the inspection of the mouth with the probe and the dental mirror.

The steps have been divided into two sub-groups, Preliminary steps (1–4) as common steps in a medical routine and Dental steps (5–8), peculiar steps expected for a dental examination.
b.The assessment of the overall level of cooperation according to the Frankl Behavioral Rating Scale, which classifies children’s behaviour into four degrees [25]: Totally Negative: the child refuses treatment, cries loudly, is afraid or faces any other test with extreme negativity;Negative: the child is reluctant to accept the treatment, uncooperative, a negative but not marked attitude is observed;Positive: the child cautiously accepts the treatment, is willing to listen to the dentist, despite his/her misgivings, follows the dentist’s instructions in a cooperative way;Totally Positive: the child has a good relationship with the dentist, is interested in dental procedures, laughs and is serene.

The level of cooperation was assessed by two independent paediatric dentists (LO and LP), trained by an experienced dentist (MGC). The training consisted of watching 10 videos of as many dental examinations of children with different degrees of collaboration, and then rating the child’s cooperation. In case of disagreement with the experienced dentist, the reasons were discussed until an agreement was reached. The same videos were seen a second time, one week later, and the intra/inter-examiner agreement was assessed trough the Inter Class Correlation Coefficient. A good agreement between the two operators was found (ICC = 0.87; 95% CI = 0.79/0.89). The mean difference between the two observers was −0.02 0.27; 95% CI = −0.06/0.05.

The secondary outcome was the parent’s judgment on the effectiveness of the visual aids provided and the overall judgment on the child’s dental experience. This information was obtained through two close-ended questions. The questions were asked to the parents by a dentist (SM) at the end of the examination (Appendix A).

At the enrolment, parents had to supply all the medical records relating to their child’s ASDs diagnosis according to the Diagnostic and Statistical Manual of Mental Disorders, 5th edition, concerning ASDs severity and verbal fluency.

Three different paediatric dentists (SM, LO, LP) performed the visit with different roles: two of them (LO, LP) performed the examination and gave their judgment on the collaboration, while the last one (SM) collected data on the steps achieved and interviewed the parents.

Before the visit, SM interviewed the parents following a questionnaire developed for this purpose (Appendix A) to collect the following information:-previous use of visual pedagogy;-use of the visual aid sent, frequency and with whom (parent and/or therapist);-use of visual aids other than those provided.

All dental examinations were conducted by LO and LP, blinded to the patient group assignment. The procedure was conducted as follows: a dentist (LO) went to the waiting room, called the patient by name, introduced himself and invited the patient and parents to follow him to the operating room; he then conducted the visit, giving indications to the patient and introducing each instrument. During the visit, the only methods used to implement subjects’ cooperation were the Tell-Show-Do method and the Positive reinforcement.

Dentin caries lesions in primary and permanent teeth were recorded according to the dmft/DMFT index. When oral inspection was not possible, caries data were collected outside the study protocol using advanced behaviour management techniques.

At the end of the visit, the two dentists independently rated the patient’s cooperation according to the Frankl Scale.

### 2.3. Data Analysis

The required sample size was powered by a standardised percentage difference of 0.5 at the Preliminary steps and Dental steps post-randomization for the primary outcome. The Alpha error was set at 0.05, the power  =  0.90, and allowances of 22% attrition due to loss of participants at follow-up and corrections in effective sample size were attributable to clustering effects of individuals. On this basis, the sample size was 31 participants per group (62 children in total). An independent operator (SM) randomized the subjects into the two groups. To avoid unbalanced groups, a stratified blinded randomization was performed. An age stratified procedure was applied (children aged between three and five years; children aged between six and eight years; children aged between nine and 14 years). Children were entered into an age-stratified data sheet (Microsoft Excel^®^) and using a systematic sampling, each subject was randomized at fixed intervals of four until the allocation was completed.

Two authors (GC and CS) blinded to the type of visual aid used conducted the statistical analysis, and the visual aid code was kept sealed until the end of the analysis.

All data obtained were retrieved in a Microsoft Office Excel^®^ sheet for data processing. In order to check the homogeneity of the two groups with regard to vital statistics (age, sex, severity of autism, verbal fluency, previous use of visual pedagogy tools and caries status) the Anova One-way or the Chi-square/Fisher’s exact test were used.

The data from the questionnaire were properly analysed using the Chi-square/Fisher’s exact test.

The differences between the two groups with regard to the number of steps achieved in Preliminary and Dental parts were analysed using the Chi-square test.

A predictive model for the attainment of the Preliminary or Dental steps was run using a multivariate logistic regression procedure in order to estimate the ORs of the steps attained and the covariates (severity of autism, verbal fluency, age range and number of caries lesions). For the Dental steps achieved, the Preliminary steps partially attained were also used as a covariate. During the multivariate analysis, the process of assessing the models demonstrated that verbal fluency and autism severity were inversely proportional (effect modifier). A new dummy variable was therefore created as the sum of the two variables.

Intraclass correlation coefficient (ICC) and 95% confidence intervals were calculated on the basis of a two-way, single-operator random-effects model to assess the agreement between the two paediatric dentists for calibration and for the assessment of the cooperation with the dental examination. ICC values less than 0.5, between 0.5 and 0.75, between 0.75 and 0.9, and greater than 0.9 were considered as poor, moderate, good, and excellent reliability, respectively. A *p*-value < 0.05 was considered statistically significant.

## 3. Results

One hundred and thirteen subjects, whose parents agreed to participate, were randomly assigned to either *Vg* (*n* = 57) or *Pg* (*n* = 56).

Eleven subjects assigned to the *Vg* and nine to the other group missed their appointment for the examination and were excluded. Nine subjects (five in the *Vg* and four in the *Pg*) were excluded after the pre-examination interview for the following reasons: two children had not received a diagnosis of autism, one parent had modified the visual aid and six subjects had not been prepared with the visual aid for different reasons (i.e., lack of time, parents considered the aid unnecessary for their child). The vital statistics (age, sex, autism severity, verbal fluency, previous use of visual pedagogy tools and caries status) of the subjects initially allocated and those actually included showed no statistically significant differences (*p* > 0.9) (*data not shown in table*). It was therefore decided to take into consideration for all statistical evaluations only the group of subjects who actually used the visual aids provided. The randomization process is summarized in Figure 1.

From the total sample of 84 subjects (41 in *Vg* and 43 *Pg*), the majority (*n* = 72, 85.71%) were males. Table 1 summarizes the vital statistics of the two groups.

The results of the parents’ interview before the examination are summarized in Table 2.

No statistically significant differences were found in terms of time of use (*p* = 0.55), people who prepared the child (*p* = 0.11) and use of other visual aids provided by parents and/or therapists in addition to those provided (*p* = 0.17).

The children completed 7.98 ± 3.37 steps in *Vg* and 6.53 ± 3.51 steps in *Pg* (*p* = 0.06).

Table 3 summarizes the achievement of each step by subjects in the two groups. More children in the Video group achieved each step, but the differences between groups were not statistically significant. “Lie down on the chair” and “Mouth inspection with instruments” showed the lowest number of children able to complete them in both groups. Considering the first four steps (Preliminary steps), a statistically significant difference (*p* = 0.04) was found between the two groups when children were divided into those who successfully achieved all steps and those who only completed some or none.

Data analysis showed that verbal fluency and autism severity had a confounding effect, so a dummy variable was created (Table 4).

In multivariate analysis (Table 5), the video aid was statistically significantly associated with the achievement of the Preliminary steps (*p* = 0.04), while the dummy was statistically associated with both Preliminary and Dental steps (*p* < 0.01).

Considering the cooperative grade as positive (scores of 3 or 4 on the Frankl Behavioral Scale) or negative (scores 1 or 2), there was good agreement between the two pediatric dentists (ICC = 0.83; 95% CI = 0.77/0.89).

The percentage of cooperative children (those who scored 3 or 4) was higher in *Vg*; however, no statistically significant differences were observed between groups according to the rates of both dentists (Table 6).

With regard to parents’ opinions, data shows that 97.56% of the parents from the Video group and 88.37% from the Photo group considered the dental visit a positive experience for the child. Most of the parents considered the visual aid provided “very useful” or “quite useful” (78.05% in the Video group and 76.74% in the Photo group); only one parent in the Video group found the visual aid “not at all useful”. No statistically significant differences were found between the parents’ opinions on both the dental visit experience (*p* = 0.20) and the usefulness of the visual aid (*p* = 0.78).

## 4. Discussion

This study was planned and carried out to assess the effectiveness of two visual aids to prepare children with ASDs for their first dental visit. The main outcome entailed the completing of the dental examination divided into four Preliminary steps and four Dental steps, demonstrating that the Video aid was as effective as the Photo aid in increasing the collaboration of children with ASDs. The Preliminary steps involve actions common to other medical environments, instead of Dental steps which include more invasive actions such as mouth manipulation. The Video aid was statistically significantly more effective in achieving the Preliminary steps, most likely due to the strong attraction of digital technology by the subjects with ASDs. Video modelling has proven to be an effective and efficient technique for teaching children with ASDs several behaviours in terms of speaking, socialization, comprehension and self-help skills [26,27]. Nevertheless, the comparison between *Vg* and *Pg* was not statistically significant for the Dental steps.

Autism Severity and Verbal fluency play an important role in the achievement of both Preliminary and Dental steps. The dentists who performed the examinations evaluated the cooperation level of children in both groups in the same way. The parents of both groups were satisfied with the visual aids and dental experience of their children.

The scientific effectiveness of visual pedagogy for subjects with ASDs in the dental field [17] appears to be affected by several factors such as ASDs severity, previous use of visual pedagogy, previous dental experience and oral health status. In the present trial, a preliminary control of cofounding factors was performed, showing no statistically significant differences in the two groups.

Two different methods were used to measure patient cooperation: an objective one, the steps achieved by the child, and a subjective one, the Frankl Behavioral collaboration Scale.

The analysis of the steps achieved shows that simple actions, such as lowering the back on the chair or tolerating the light on the face, are important sources of stress for children with ASDs. Other steps which produce less stress such as opening the mouth, were achieved by those who refused the previous steps, allowing dentists to inspect the oral cavity in suboptimal conditions. As for the level of cooperation assessed by the operators, it was higher in the Video group, although in both groups almost half of the sample was rated negatively or totally negative. These findings strengthen the concept that children with ASDs may not only need a unique behavioural technique but also different tailored approaches, such as the use of sunglasses, to tolerate intense sensory stimuli. Together with visual aids, these approaches could contribute to the success of the visit.

### Limitation of the Study

Due to the characteristics of this study, it was not possible to blind subjects and parents to the visual aid assigned, but dentists collecting the data and performing the examinations and those performing the data analysis were blinded to the visual aid used.

For ethical reasons, no control group was used, as it was not considered appropriate to include a group of subjects without any visual aid provided, making them totally unprepared for the dental experience [24,28,29].

Although both tools used were effective in preparing the children for the dental environment, the high number of non-respondents may have influenced the results of the trial. Additional strategies may be required to improve the children’s level of cooperation in order to fully achieve the task.

The role of parents in this protocol was crucial. Children in both groups used the visual aid mainly with their parents. This highlights the key role of the family in the upbringing of the child, who must be continuously stimulated to deal with daily routine activities and special events like medical examinations. However, the data show a large variability in the frequency of visual aid use, probably because not all parents understood how visual aid could improve the patient’s cooperation. Despite this wide variability, most parents gave a positive judgment of both their child’s dental experience and the usefulness of the visual aid, including those parents whose children offered little or no cooperation with the procedure. It is possible to speculate that many uncooperative subjects at the first examination may become more cooperative after a few repeated follow up appointments. This is the case of the desensitization technique, where even a negative or partially negative first experience may contribute to increasing the child’s cooperation in subsequent assessments [22,24]. Desensitization appears to be a successful approach to provide dental care for children with ASDs, especially for those who are able to socially engage with clinicians and caregivers and perform basic self-care [30]. Furthermore, the effectiveness of the visual aids provided has only been investigated for the performance of the first oral examination, but this does not mean that visual aids cannot also be used effectively to prepare children with ASDs for dental cares. Although visual aids customized for different preventive/therapeutic procedures seem to be effective in preparing children [16], further investigations are needed to clarify their role in invasive dental procedures.

## 5. Conclusions

The results of this study emphasise that behavioural intervention through visual aids should be used as strategy to prepare subjects with ASDs for the dental examination with a good impact on both children’s cooperation and family satisfaction. However, for some children with ASDs, additional aids are required, especially for those with a greater disability.

## Figures and Tables

**Figure 1 children-09-00729-f001:**
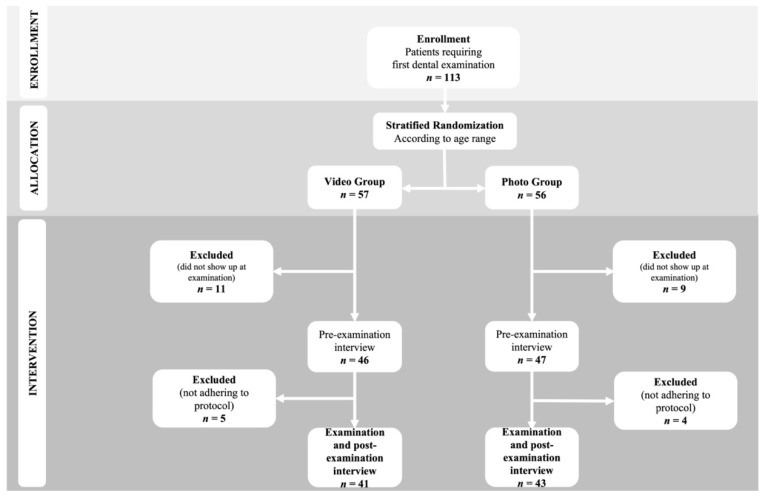
Consort flow diagram of the study.

**Table 1 children-09-00729-t001:** Vital statistics including age, sex, autism severity, verbal fluency, previous use of visual pedagogy tools and caries status of the two groups.

	Video Group	Photo Group	Total	*p*-Value
**Age**	**Mean ± SD (range)**			
	7.51 ± 2.53 (4–15)	7.56 ± 2.32 (3–13)	7.54 ± 2.42 (3–15)	0.93 ^c^
	***n*** **(%)**			
3–5 yy	9 (21.95)	9 (20.93)	18 (21.43)	0.95 ^a^
6–8 yy	19 (46.34)	19 (44.19)	38 (45.24)	
>9 yy	13 (31.71)	15 (34.88)	28 (33.33)	
Total	41 (100.00)	43 (100.00)	84 (100.00)	
**Sex**	***n*** **(%)**			
Males	34 (82.93)	38 (88.37)	72 (85.71)	0.47 ^b^
Females	7 (17.07)	5 (11.63)	12 (14.29)	
Total	41 (100.00)	43 (100.00)	84 (100.00)	
**Autism severity**	***n*** **(%)**			
Grade I	11 (26.83)	10 (23.26)	21 (25.00)	0.80 ^a^
Grade II	12 (29.27)	11 (25.58)	23 (27.38)	
Grade III	18 (43.90)	22 (51.16)	40 (47.62)	
Total	41 (100.00)	43 (100.00)	84 (100.00)	
**Verbal fluency**	***n*** **(%)**			
Non verbal	16 (39.02)	20 (46.51)	36 (42.86)	0.75 ^a^
Non fluent	11 (26.83)	11 (25.58)	22 (26.19)	
Fluent	14 (34.15)	12 (27.91)	26 (30.95)	
Total	41 (100.00)	43 (100.00)	84 (100.00)	
**Previous use of visual pedagogy**	***n*** **(%)**			
Yes	26 (63.41)	30 (69.77)	56 (66.27)	0.64 ^a^
No	15 (36.59)	13 (30.23)	28 (33.73)	
Total	41 (100.00)	43 (100.00)	84 (100.00)	
**Caries lesions**	**Mean ± SD (range)**			
	2.73 ± 2.93	3.00 ± 2.84	2.87 ± 2.89	0.67 ^c^
	*n* (%)			
Caries free	15 (36.59)	11 (25.58)	26 (30.95)	0.44 ^a^
1–3 lesions	10 (24.39)	15 (34.89)	25 (29.76)	
>4 lesions	16 (39.03)	17 (39.53)	33 (39.28)	
Total	41 (100.00)	43 (100.00)	84 (100.00)	

^a^ Chi-square test; ^b^ 2-tailed Fisher’s exact test; ^c^ One-way Anova.

**Table 2 children-09-00729-t002:** Findings of the parents’ interview before the visit.

	Video Group *n* (%)	Photo Group *n* (%)	Total	*p*-Value
**Frequency of use of the visual aid**				
1 time per week	10 (24.40)	6 (13.96)	16 (19.04)	0.55 ^a^
3–4 times per week	9 (21.95)	14 (32.56)	23 (27.38)	
1 time per day	16 (39.02)	17 (39.53)	33 (39.29)	
>1 time per day	6 (14.63)	6 (13.95)	12 (14.29)	
Total	41 (100.00)	43 (100.00)	86 (100.00)	
**Administrator of the visual aid**				
Parents	37 (90.24)	32 (74.42)	69 (82.14)	0.11 ^b^
Therapist	0 (0.00)	1 (2.32)	1 (1.19)	
Both parents and therapist	4 (9.76)	10 (23.26)	14 (16.67)	
Total	41 (100.00)	43 (100.00)	86 (100.00)	
**Use of additional visual aids**				
Yes	11 (26.83)	18 (41.86)	18 (41.86)	0.17 ^a^
No	30 (73.17)	25 (58.14)	25 (58.14)	
Total	41 (100.00)	43 (100.00)	43 (100.00)	

^a^ Chi-square test; ^b^ Fisher’s exact test.

**Table 3 children-09-00729-t003:** Number of subjects who overcame each step and divided in Preliminary and Dental steps partially or not achieved and fully achieved in the two groups.

	Video Group (*n* 41)	Photo Group (*n* 43)	*p*-Value
**Steps achieved**	*n* (%)		
1. Entering the room	41 (100.00)	41 (95.35)	0.49
2. Sitting on the dental chair	35 (85.37)	35 (81.40)	0.77
3. Laying in the dental chair	33 (80.49)	29 (67.44)	0.21
4. Tolerating light on the face	35 (85.37)	29 (67.44)	0.07
5. Opening the mouth	34 (82.93)	32 (74.42)	0.42
6. Mouth manipulation with gloves	32 (78.05)	29 (67.44)	0.33
7. Mouth inspection with a dental mirror	29 (70.73)	24 (55.81)	0.18
8. Mouth inspection with a dental probe	24 (58.54)	16 (37.21)	0.08
**Preliminary steps (1–4)**			
Not or Partially achieved	8 (19.51)	17 (39.53)	
Fully achieved	33 (80.49)	26 (60.47)	
	χ^2^_(3)_ = 4.03 *p* = 0.04		
**Dental steps (5–8)**			
Not or Partially achieved	17 (41.46)	27 (62.79)	
Fully achieved	24 (58.54)	16 (37.21)	
	χ^2^_(3)_ = 3.83 *p* = 0.05		

**Table 4 children-09-00729-t004:** Association between verbal fluency and autism severity.

Verbal Fluency	Autism Severity 1 *n* (%)	Autism Severity 2 *n* (%)	Autism Severity 3 *n* (%)
Fluent	14 (53.85)	12 (46.15)	0 (0.00)
Non Fluent	7 (31.82)	8 (36.36)	7 (31.82)
Non verbal	0 (0.00)	3 (8.33)	33 (91.67)
χ^2^_(8)_ = 55.03 *p* < 0.01			

**Table 5 children-09-00729-t005:** Logistic estimates of achievements of preliminary or dental steps (multivariate logistic regression).

**Preliminary steps (1–4)** N of observation = 84 Log likelihood = −41.86 *p* < 0.01			
**Covariates**	**OR (SE)**	***p*-value**	**95% CI**
Visual support (*Vg*)	0.33 (0.18)	0.04	0.11/0.98
Autism/Verbal (*low severity and fluent verbal*)	0.54 (0.11)	<0.01	0.36/0.79
Age groups	1.41 (0.51)	0.35	0.69/2.87
Constant	43.67 (61.65)	<0.01	2.75/694.64
**Dental steps (5–8)** N of observation = 84 Log likelihood = −32.59 *p* < 0.01			
**Covariates**	**OR (SE)**	***p*-value**	**95% CI**
Visual support (*Vg*)	0.50 (0.32)	0.28	0.14/1.73
Autism/Verbal (*low severity and fluent verbal*)	0.44 (0.10)	<0.01	0.28/0.68
Preliminary steps (partially achieved)	28.10 (31.41)	<0.01	3.14/251.33
Age groups	1.26 (0.31)	0.34	0.78/2.4
Constant	1.88 (3.12)	0.70	0.07/48.37

**Table 6 children-09-00729-t006:** Collaboration scores according to the Frankl Behavioral Scale in both groups expressed by the two dentists (LO and LP) who performed the oral examination.

	Video Group	Photo Group	*p*-Value
	*n* (%)	*n* (%)	
	**Visit**		
**Dentist (LO)**	*n* (%)		
Totally negative	5 (12.20)	13 (30.23)	0.24 ^b^
Negative	13 (31.70)	10 (23.26)	
Positive	14 (34.15)	11 (25.58)	
Totally positive	9 (21.95)	9 (20.93)	
Total	41 (100.00)	43 (100.00)	
**Dentist (LP)**	*n* (%)		
Totally negative	7 (17.07)	12 (27.91)	0.48 ^a^
Negative	11 (26.83)	10 (23.26)	
Positive	13 (31.71)	15 (34.88)	
Totally positive	10 (24.39)	6 (13.95)	
Total	41 (100.00)	43 (100.00)	

^a^ Chi-square test; ^b^ Fisher’s exact test.

## Data Availability

Data supporting reported results can be found in Appendix A.

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
