# Peer review of "Use of Visual Pedagogy to Help Children with ASDs Facing the First Dental Examination: A Randomized Controlled Trial"

_children, 2022, doi:10.3390/children9050729_

Round 1
Reviewer 1 Report
This is a very interesting and well-conducted study. However, I recommend to clarify some items listed below.
page 2, last line: what is meant by "selected at intervals of four"?
page 3, step 3: what was the definition of reclined dental chair? In my experience many children with any type of impairment do not like the movement of the back of the dental chair. Many patients do not accept an angle more than 45° and this only when the movement back is carried out in small steps. So here more information is requrired.
step 4: please explain more precisely "turns on the light of the dental chair": was this done by lighting at the same time the face or was this done without lighting the face?
term "medical steps": I am not happy with this term because the dental chair is something very special and cannot be compared with a couch that may be present in the cabinet of a pediatrician. I recommend to chose another term. Steps 1 to 4 refer to the situation without involving the mouth and the oral cavity while in this happens in steps 5 bo 8.
page 4, line 200: what do you mean by saying "verbal fluency and autism severity were related" Is there not an inverse relationship?
Sentence above table 6: Please reword this sentence because the figures 56.1 and 56.12 are nearly the same! It is very clear that there will be no statistically significant difference.
Discussion: I strongly to recommend to discuss also that this type of preparation was not successful for about 50% of the PG and about 43% of the VG (see table 6). The reader should get an impression whether the invervention of this study is successful because otherwise the proportion of non-cooparating children would have been much higher or not.
Author Response
Please, see the attached file

Reviewer 2 Report
The topic of this manuscript is interesting as use of both photo-based and video-based visual pedagogy techniques are commonly utilized as aids to improve cooperation and experience of children with ASD during dental care.
English/Language choices:
- There are multiple cases of grammar errors that make some of the manuscript challenging to read. Suggest editing by a native English speaker.
- Line 46: I am unclear with the meaning intended by use of the term ‘deadlock’ here. Please clarify.
- Please reconsider language choices regarding autism. There is a current discrepancy between person-first and identity-first language (e.g., person with ASD and autistic person). The autism community seems to be favoring identity-first language, although the authors can make their own final choice. That said, “ASDs children” or “ASDs subjects” need to be changed throughout the manuscript.
Abstract: adequate
Introduction
- Overall, clear and concise description of the literature and rationale for this research project.
- There are a few statements that read a bit too broad and/or causal for my comfort. For example:
- Lines 39-41 “ASDs have not a direct outcome on oral health but are associated with severe difficulties in….”
- Lines 43-46 “…compromised communication skills might generate behavioural problems.” Although this statement is technically true, the literature suggests that behavioral problems are much more complex and multifactorial than just one challenge (e.g., communication skills) causing behavioral issues during care. Suggest rephrasing to something like “compromised communication skills have been linked to/associated with behavioral problems” (and then including a reference; e.g., Marshall et al., 2007).
- Lines 46-48 This description of ASD feels to broad; not all individuals with ASD present this way. In addition, although ADHD is often comorbid with ASD, ‘hyperactivity with attention deficit’ is not a key component of an ASD diagnosis.
- Some reference-related improvements could be made. For example:
- Lines 49-51 missing reference
- Lines 55-58: Ref #10 is from 1999. Also, this reference is incorrect – instead of U, K; AJ, N should be Klein, U.; Nowak, A.J.
- Lines 65-66 missing reference; could avoid needing a review reference here by changing language from “main cognitive behavioural therapy…” to “common cognitive…”.
- Lines 73-74: seems like ref #17 is in the incorrect location as it does not focus on the use of video-modeling in dentistry, but just video-modeling in general.
Methods
- Good description of randomization procedures.
- Frankl Scale reference is incorrectly referenced (#23) – Not Child, S.F.-J.D., but should be Frankl, S.N. Also a ‘U’ and 1962 in author location? Please double check all references for accuracy.
- Concerned/curious how medical records allowed for “a correct classification of ASDs severity and verbal fluency” (lines 154-156). Please add additional detail regarding exactly what this data was, what met criteria for inclusion, and how this data was collected from the medical records. Also recommend defining and/or referencing the three Severity levels for ASD here (as they are reported later in Table 1).
- Suggest merging the two (currently separate) sections about the secondary outcome – lines 150-153 & 178-182.
- Is S4 the only guide for the “interview”? If so, I wouldn’t call this an interview.
- Would appreciate more information regarding the secondary outcome of the parents’ judgment on the efficacy of the visual aids and child’s dental experience. I understand this was completed via interview, but did it follow a semi-structured guide? What types of questions were answered? How long did these interviews last? Did all participants engage in the interviews? How was the data recorded (notes vs. recordings and later review of interviews)? Etc.
Results
- Table 2: please add more detail in the manuscript text regarding the “use of additional visual aids”.
- Table 4: As verbal fluency is a defining factor of ASD level severity level (as defined by the DSM-5), I don’t understand the purpose of this table.
Discussion
- Line 294 “probably due to the high attraction of digital technology by ASDs subjects”: could you add further discussion regarding this assumption, or references to support this statement?
- Lines 335-337: consider adding some conversation surrounding desensitization techniques here.
Author Response
Please, see the attached file
